# *Lactobacillus salivarius* SNK-6 Activates Intestinal Mucosal Immune System by Regulating Cecal Microbial Community Structure in Laying Hens

**DOI:** 10.3390/microorganisms10071469

**Published:** 2022-07-20

**Authors:** Yuchen Liu, Lianhong Li, Huaxiang Yan, Zhonghua Ning, Zhong Wang

**Affiliations:** 1National Engineering Laboratory for Animal Breeding and Key Laboratory of Animal Genetics, Breeding and Reproduction, Ministry of Agriculture and Rural Affairs, College of Animal Science and Technology, China Agricultural University, Beijing 100193, China; cauliuyuchen@163.com; 2State Key Laboratory of Animal Nutrition, College of Animal Science and Technology, China Agricultural University, Beijing 100193, China; 18810871557@163.com; 3Institute of Animal Husbandry and Veterinary Science, Shanghai Academy of Agricultural Science, Shanghai 201106, China; yanpoultry@163.com

**Keywords:** *lactobacillus salivarius*, hens, immunity, microbiome

## Abstract

The production performance and disease resistance of laying hens decrease obviously with age. This study aimed to investigate the effects of supplementary *Lactobacillus salivarius* (*L. salivarius*) *SNK-6* on laying performance, the immune-related gene expression in cecal tonsil, and the cecal microbial composition of laying hens. Here, 384 Xinyang black commercial hens (55 weeks old) were randomly allocated to three groups under the same husbandry and dietary regimes: basal diet (Con), the low *L. salivarius SNK-6* group (T1: 1.0 × 10^6^ CFU/g), and the high *L. salivarius SNK-6* group (T2: 1.0 × 10^7^ CFU/g). The results showed that the feed intake and broken-egg rate in the T1 group were significantly higher than the Con group (*p* < 0.05). Meanwhile, expressions of intestinal mucosal immune-related genes were significantly upregulated. The 16S rRNA gene sequencing indicated that supplementary *L. salivarius SNK-6* had no significant difference in α -diversity and only displayed a trend difference in the β-diversity of cecal microbiota (*p* = 0.07). LEfSe and random forest were further used to identify bacteria family *Enterobacteriaceae*, order *RF39*, genera *Ochrobactrum*, and *Eubacterium* as biomarkers between the Con and T1 groups. Genera *Ochrobactrum*, which had high relative abundance and nodal degree in the T1 and T2 groups, showed a significant positive correlation with the expression of TLR-6, IL-10, MHC-II, and CD40 in cecal tonsils and might play a critical role in activating the host intestinal mucosal immune responses. Overall, dietary supplementary *L. salivarius SNK-6* can display an immunomodulatory function, possibly by regulating cecal microbial composition. However, the changes in immune responses may be at the expenditure of corresponding production performance, which needs to be weighed up in practical application.

## 1. Introduction

With an increase in large-scale rearing, especially under high-density conditions, the laying performance, antioxidant capacity, and disease resistance decrease rapidly after the peak-laying period [1,2]. Moreover, hens are often accompanied by microflora imbalance [3] and the development of chronic inflammation during aging [4,5]. These changes lead to lowered immunity, increased pathogen susceptibility, and significant economic losses in aged hens. However, antibiotics have been used to improve the egg-laying rate and control bacterial infections caused by pathogenic *Escherichia coli*, *Clostridium perfringens*, and *Salmonella* after the peak-laying period. Still, many countries globally have banned or limited the antibiotics used in laying stages due to antibiotic resistance, environmental pollution, and egg safety [6]. Therefore, alternative strategies to antibiotics for improving the production performance, immune function, and enhanced disease resistance of hens during the late-laying period are significant in prolonging the laying cycle and service life [7].

*Lactobacillus salivarius**SNK-6* is a Gram-positive bacilli isolated from the gut of a healthy chicken. Several common strains of *Lactobacillus* are well-recognized as probiotics, which can colonize and grow on the intestinal surface due to their capacity for intestinal dietary adsorption and the production of bacteriocins [8,9]. Many previous studies have shown that *L. salivarius* can increase the body weight gain, body weight, and feed conversion ratio (FCR), while simultaneously reducing the contents of the total cholesterol, low-density lipoprotein, and triglyceride in the blood [10,11,12,13,14]. *Lactobacillus salivarius* displays high acid and bile salt tolerance and can facilitate its colonization by producing antimicrobial peptides and bacteriocin in the intestine, hence, adjusting the intestinal microflora and modulating the immune system [15,16]. Additionally, *L. salivarius* can reduce the organ damage caused by the infection of Mycoplasma gallisepticum (MG), *Campylobacter*, and *Escherichia coli O78*, augmenting the immune responses after infectious bursal disease virus (IBDV) vaccination, enhancing the production of specific antibodies and stimulating lymphocyte proliferation [8,17,18]. Moreover, studies have shown that *L. salivarius* or its cellular components can effectively activate the intestinal mucosal immune response, increasing intraepithelial lymphocytes, and IgA-producing cells and improving the expression and release of interleukin (IL-6), beta-defensins-2, and TLR-2 [19,20]. Meanwhile, in vitro assays have indicated that *L. salivarius* could promote naive T-cell differentiation (Th1) and participate in immunomodulatory responses [21].

However, the effect of *L. salivarius* on the laying performance, intestinal flora, and the relationship between microflora and host immune responses in hens have not been sufficiently elucidated in hens. This study aimed to explore whether the supplementation of *L. salivarius* could improve the production performance and immunity and further reveal the immunoregulatory mechanisms by analyzing the gut microbiota of aged hens.

## 2. Materials and Methods

### 2.1. Feed Preparation

The *L. salivarius SNK-6* additive, containing 1.0 × 10^11^ CFU/g, was provided by Professor Yan Huaxiang from the Shanghai Academy of Agricultural Sciences with the preservation number CCTCC NO: m2018044. To ensure the number and viability of *L. salivarius SNK-6*, the diets were formulated every 7 days. We first calculated the exact amount of bacteria and basal diet required to produce the trial diet, then diluted the prepared bacteria with a relatively small amount of basal diet and mixed them with the rest of the feed. Finally, diets with low dose *L. salivarius SNK-6* group (1.0 × 10^6^ CFU/g of diet) and high dose *L. salivarius SNK-6* group (1.0 × 10^7^ CFU/g of diet) were obtained.

### 2.2. Birds’ Diets and Management

All manipulation and experimental procedures involving animals were performed according to the principles of the Animal Care and Use Committee of China Agricultural University, Beijing (permit number: AW30601202-1-1). The animal welfare committee approved the bird management and handling procedures. A total of 384 55-week-old Xinyang black commercial hens with similar production performance and body weight were randomly divided into three groups: the control group fed with basal diet (Con), the low dose group (T1: 1.0 × 10^6^ CFU/g), and the high dose group (T2: 1.0 × 10^7^ CFU/g) with eight replicates of 16 hens in each (four cages in each replicate, four hens in each coop, 40 cm wide, 62 cm long, and 45 cm high). A corn-soybean meal-based diet was formulated according to the NY/T-33-2004 feeding standard [22]. The compositions and primary nutrient contents of the basal diet are given in Table 1. The experiment was performed at laying hen farms in Fengxian District, Shanghai. The trial lasted 9 weeks including a 1-week pre-experiment and an 8-week formal experiment. The chicken house temperature was maintained at 23 °C to 25 °C through an automatic environment control system from April to June, Shanghai 2019. All hens were raised in cages under a daily regimen of lighting 16 L:8 D and were allowed feed and water ad libitum throughout the trial.

### 2.3. Laying Performance Parameters

The number of eggs laid, broken eggs, abnormal eggs, egg weights, and dead chickens in each replicate were recorded daily. The feed intake for each repetition was counted every week. The average egg production rate, average egg weight, broken egg rate, abnormal egg rate, feed egg ratio, mortality, and average daily feed intake was calculated (from weeks 1 to 4, from week 5 to 8, and from weeks 1 to 8).

### 2.4. Quantitative Real-Time PCR for Measuring Immune-Related Gene Transcript Level in the Cecal Tonsil

Six hens were randomly selected per treatment at the end of the experiment (64 weeks old), and euthanasia was performed using carbon dioxide. Each hen’s cecal tonsil tissue and cecal contents were rapidly collected within 5 min and immediately frozen in liquid nitrogen until follow-up trials. Real-time quantitative RT-PCR (qRT-PCR) was performed to assay the immune-related gene expression of the cecal tonsils following the previous method [23]. Specifically, the total RNA in tissues was extracted using Trizol. The quality and concentration of RNA were determined using a micro-spectrophotometer Nano-300 (Hangzhou Allsheng Instruments Co., Ltd., Hangzhou, China) and 1% (*w*/*v*) agarose gel. High-quality RNA was reversed-transcribed into cDNA with the Primer Script RT Reagent Kit (Takara Bio, Beijing, China). The qRT-PCR amplification was performed using a 7500 fast real-time PCR system (Applied Biosystems Inc., Waltham, MA, USA) with a SYBR^®^ Premix Ex Taq™ Kit (Takara, Beijing, China). Here, GAPDH was used as the reference gene for quantitative real-time RT-PCR. The cytokines measured were IL-1β, IL-2, IL-4, IL-10, IL-12, IFN-γ, TNF-α, TLR-2, TLR-4, TLR-6, NF-κB, MyD88, MHC-II, MHC-I, CD86, CD80, CD40, CD40L, and CD28 in the cecal tonsil. The primer sequences of the genes are provided in Appendix A, and the relative expression of each target gene was calculated by the 2^−ΔΔCT^ method.

### 2.5. DNA Extraction, Amplification, and Sequencing

Total bacteria genomic DNA samples were extracted using the Soil DNA Kit (D5625-01; Omega Bioservices, Norcross, GA, USA) according to the manufacturer’s instructions. The concentration and purity of the total DNA were determined by a Nanodrop 2000 (Thermo Fisher Scientific, Waltham, MA, USA) and agarose gel electrophoresis assay. The bacterial 16S rRNA V3–V4 region was amplified using PCR barcode primers (338F: 5′-ACTCCTACGGGAGGCAGCA-3′ and 806R: 5′-GGactachVGGGTWTCTAat-3′) and a KAPA HiFi Hot Start Ready Mix PCR Kit (KAPA Biosystems, Wilmington, MA, USA). PCR products were detected by 2% agarose gel electrophoresis, and the target fragments were purified by Axygen Kits (Axygen, Union City, CA, USA). The purified PCR products were employed for library construction. Sequencing was performed using an Illumina MiSeq platform with two paired-end read cycles of 300 bases each (Illumina Inc., San Diego, CA, USA; Personalbio, Shanghai, China).

### 2.6. Bioinformatic Analyses

After sequencing, fastq files were processed with QIIME2 software (Version 2020.8) [24]. DADA2 was further used to denoise sequences and produce the amplicon sequence variant (ASV) tables [25]. A Naïve Bayes classifier was trained on a Greengenes (version 13_8) database (99% identity clusters) containing the V3–V4 region using the feature-classifier classify-sklearn function [26]. For phylogenetic diversity measures, a rooted phylogenetic tree was created with Fasttree [27]. Diversity analysis covered displays and statistics such as alpha diversity (Chao1, Observed species, Shannon, and Simpson), principal coordinate analysis (PCoA), rarefaction curves, and microbial stacked bar plots were conducted by MicrobiomeAnalyst (https://www.microbiomeanalyst.ca) (accessed on 2 November 2021), and the filtering parameters were set as the default [28]. The linear discriminant analysis (LDA) effect size (LEfSe) (http://huttenhower.org/galaxy) (accessed on 23 October 2021) was used to identify the taxa that explain the differences in microbial communities between the Con vs. T1 and Con vs. T2 groups [29]. Taxa with an |LDA score| > 2.0 and a *p*-value < 0.05 were significant in this study. The microbial co-occurrence network analysis was inferred by the CCLasso algorithm based on the genus level to elucidate the characteristics of the taxon–taxon interactions in each group by MetagenoNets with default parameters and FDR values of *p* > 0.05 and r < 0.6 were treated with 0 [30]. The topological properties of the network nodes were analyzed and standardized by z-transformation before visualization. Then, the network correlation diagram was visualized by Gephi software version 0.9.2 (The Gephi Consortium, Paris, France). Next, random forest analysis was conducted at the genus level to identify particular genera from the available microbiome data. The significance of each predictor on the response variables was assessed by an R package “rfPermute” (1000 iterations) for permutated random forests, and values of *p* < 0.05 were considered significant [31]. Subsequently, we performed Boruta feature selection using the R package “Boruta” [32]. The Boruta algorithm is a wrapper based on random forest classification that compares the importance of real features with “shadow attributes” with randomly shuffled values. After 1000 iterations, features with lower importance than “shadow attributes” will be iteratively removed. The genera were identified as biomarkers unless the following conditions were satisfied simultaneously. The Spearman rank correlation test was used (R package “psych”) to analyze the correlations between the signature microbial taxa (the union of significant results obtained by Random Forest and LEfSe) and the significant differentially expressed genes of the cecal tonsil. FDR adjusted *p* < 0.2 was considered statistically significant and visualized using the R package “corrplot”. The R package “ggplot2” and “ggridges” were used to complete other figures in this study.

### 2.7. Statistics Analysis

All apparent data were analyzed using SPSS 24.0 statistical software (SPSS Inc., Chicago, IL, USA). The production performance and gene expression data were analyzed with a normal distribution test and homogeneity test of variance. Student’s t-test conducted the indices that passed the test; otherwise, the Wilcoxon non-parametric test was used. The Kruskal–Wallis tested the alpha diversity metrics. The PCoA (principal coordinates analysis) was calculated for the distance matrix by the Jensen–Shannon divergence distance method, and the permutational MANOVA (PERMANOVA) test (also known as Adonis) was used to verify its significance. The apparent results were expressed as the mean and standard error. Differences at *p* < 0.05 were considered significant, whereas *p* values between 0.05 and 0.1 were interpreted as trends.

## 3. Results

### 3.1. Laying Performance

The laying performance data are shown in Appendix B. The results showed no significant differences in egg production, weight, mass, and mortality among all groups (*p* > 0.05). However, the feed intake of the T1 group was significantly increased at weeks 1–8 and 1–4 as well as the rate of broken eggs at weeks 1–4 compared with the control group (*p* < 0.05). In addition, the rate of broken eggs at weeks 1–8 and FCR at 5–8 weeks in the T1 group also exhibited a significant trend of improvement (*p* = 0.097 and *p* = 0.05, respectively).

### 3.2. Cecal Tonsil Cytokines mRNA Levels

The cytokine mRNA expression of the cecal tonsils after adding *Lactobacillus salivarius SNK-6* is shown in Appendix C. Relative to the control group, IL-1β, IL-10, IL-12, TLR-6, and MHC-II were significantly upregulated in the T1 group (*p* < 0.05); NF-κB, MyD88, CD86, CD80, CD40, CD40L, and CD28 were significantly increased in both the T1 and T2 groups (*p* < 0.05). In addition, TLR-2, TLR-4, and MHC-I had an elevated-trend response to SNK-6 supplementation (*p* = 0.096, *p* = 0.069 and *p* = 0.087 respectively), while IL-4 showed the opposite trend (*p* = 0.077).

### 3.3. Intestinal Bacterial Richness, Diversity, and Similarity

A total of 468,613 sequences were obtained from all samples, with an average of 26,034 sequences per sample after size filtering, quality control, and chimera removal. The bacterial community composition analysis resulted in 1488 unique ASVs reads (ASVs with ≥2 counts), and after data filtering, there were 773 unique ASVs for downstream analysis (20% samples of its ASV values should contain at least four counts; the last 10% ASVs of the inter-quantile range were removed). Data without filtering were used for alpha diversity, and other downstream analyses were performed by the filtered ASV table. The alpha diversity rarefaction curves (Figure 1a) reached a stable plateau under the sequencing depth, indicating a reasonable sequencing depth. The alpha diversity index indicates the richness and uniformity of the species composition. There was no significant difference in the alpha diversity between each group in this study (Figure 1b). The principal coordinate analysis (PCoA) based on the Jensen–Shannon divergence distance method showed obvious separation (Figure 1c). Principal components PC1, PC2, and PC3 explained 19.7, 16.5, and 11.1% variation, respectively (PERMANOVA, *p* = 0.07, R^2^ = 0.15).

### 3.4. Ileal Microbial Community Structure

Firmicutes, Bacteroidetes, Proteobacteria, and Actinobacteria are the dominant phyla (relative abundance > 1%) in the cecum of hens, accounting for more than 95% of the total bacterial community (Figure 2a). The addition of SNK-6 increased the relative abundance of Firmicutes (Con: 54.7%, T1: 58.2%, and T2: 57.9%) and decreased the relative abundance of Bacteroidetes (con: 35.8%, T1: 31.5%, and T2: 31.7%). In addition, Verrucomicrobia was significantly enriched in the T1 group (Con: 0.009%, T1: 2.2%, and T2: 0.07%).

Bacteroides, Ruminococcus, Oscillospira, Phascolarctobacterium, Desulfovibrio, Dorea, Faecalibacterium, Lactobacillus, Parabacteroides, Blautia, and Butyricicoccus were the dominant genera of the three groups (Figure 2b) (relative abundance > 1%). Interestingly, we found that Akkermansia was significantly enriched in the T1 group (Con: 0.009%, T1: 2.2%, and T2: 0.07%), while Megamonas was inhibited considerably in the T2 group (Con: 1.16%, T1: 1.19% and T2: 0.26%).

### 3.5. Key Microbial Identification

Linear discriminant analysis (LDA) effect size (LEfSe) analysis (with LDA scores > 2.0) was performed to determine the specific taxa responsible for the differences among the three groups (Figure 3a,b). Compared to the Con and T1 groups, the bacteria family Enterobacteriaceae and order RF39 were significantly enriched in the Con group. In a comparison between the Con and T2 groups, the bacteria family Enterobacteriaceae, genera Megamonas, species Bacteroides barnesiae, and Bacteroides Plebeius were significantly enriched in the Con group. Meanwhile, we found that Enterobacteriaceae at the family level was specifically enriched in the Con group. The genera Megamonas, species Bacteroides barnesiae, and Bacteroides plebeius were specifically suppressed, while Desulfovibrio D168 at the species level and SHA-98 at the order level were explicitly enriched in the T2 group (Figure 3c).

Taxa identified by LEfSe came from different levels. To further mine the potentially important bacteria at the genus level, we used a random forest classifier to define reliable biomarkers of the gut microbial response to SNK-6 (Figure 4a,b). In comparisons of the Con and T1 groups, the genera Ochrobactrum and [Eubacterium] were identified as a biomarker to distinguish the two groups. In comparisons of the Con and T2 groups, the genera Bacteroides, Alistipes, and Megamonas were identified as the critical bacteria.

### 3.6. Co-Occurrence Patterns of Microbial Communities

The analysis of microbial co-occurrence patterns can provide valuable insights into functional microbial communities. Figure 5a depicts the microbial co-occurrence network of groups Con, T1, and T2. The bacterial network in the Con, T1, and T2 groups contained connected nodes (31, 51, 49, respectively) and edges (64, 258, 171, respectively), with an average node degree (4.1, 10.1, 7.0, respectively). We found that adding the bacteria SNK-6 increased the number of interacting connected nodes, and 19 genera were specifically activated in the T1 and T2 networks (Figure 5b). The network complexity in the T1 and T2 groups was significantly higher than that of the Con group, indicating tighter interactions with microbial communities after adding SNK-6. We next calculated the topological network properties of each connected node to describe the complex patterns of correlations between the microbial genera (Figure 5c). The results showed that the top three genera node connectivity in the Con, T1, and T2 groups were [Con: ([Ruminococcus] (degree = 13); Faecalibacterium (degree = 9); Desulfovibrio (degree = 9)), T1: (Dorea (degree = 21); [Eubacterium] (degree = 21); Bacteroides (degree = 20)), T2: (Blautia (degree = 11); Butyricicoccus (degree = 11); CC_115 (degree = 10)), respectively]. Finally, we focused on the topological attributes of the signature genera identified by LEfSe and random forest. Megamonas (Con: 1.16%, T1: 1.19%, and T2: 0.26%) was a genus with decreased relative abundance in the T2 group, and the node degree in the Con, T1, and T2 groups were 3, 4, and 8, respectively. It also had a very high betweenness of 0.59 (No. 3) in the Con group. Ochrobactrum (Con: 0.009%, T1: 0.03%, and T2: 0.02%) was a genus with a higher relative abundance in the T1 and T2 groups, and the node degree in the Con, T1, and T2 groups were 1, 19, and 8, respectively. Eubacterium (Con: 0.37%, T1: 0.20%, and T2: 0.23%) was a genus with higher relative abundance in the Con group, and the node degree in the Con, T1, and T2 groups was 2, 21, and 8, respectively. Finally, the Bacteroides (Con: 14.88%, T1: 11.83%, and T2: 9.98%) were bacterium with high relative abundance in the Con group, and the node degree in the Con, T1, and T2 groups was 3, 20, and 1, respectively.

### 3.7. Correlation among Signature Taxa and Differential Genes in Cecal Tonsil

It is vital to construct a network between the signature microbiota and host gene expression to understand better how the intestinal host–microbial relationship regulates host defense and inflammation (Figure 6). Results of the Spearman’s correlations showed that microbiota significantly enriched in the Con group (order RF39, family Enterobacteriaceae, genera Bacteroides, and species Bacteroides plebeians) indicated a markedly negative regulation effect on the expression of cecal tonsil immune-related genes (e.g., TLR-6, MyD88, NF-κB, MHC-II, CD80, CD40, and CD28). In contrast, the relative abundance of these microbial taxa was lower in groups T1 and T2. Notably, the microbial taxa significantly enriched in the T1 group (genera Ochrobactrum, Alistipes, and species Bacteroides Barnesiae) showed significant positive correlations with cecal tonsil immune-related genes (e.g., TLR-6, IL-10, IL-12, MHC-II, CD40). Interestingly, the order SHA-98 and species Desulfovibrio D168, markedly enriched in the T2 group, showed a significant negative regulation effect on IL-12 expression in the cecal tonsil. In addition, we found that the relative abundance of the genera Ochrobactrum and Alistipes and species Bacteroides Barnesiae, which promote the expression of immune-related genes, was slightly downregulated in the T2 compared with the T1 group. It may partly explain the weaker immune-related gene expression in the T2 group compared to the T1 group.

## 4. Discussion

Aging is a standard and complex biological process. The laying performance and immune response of hens rapidly decrease during the late egg-laying period with age [1]. Bacteria are widely distributed in the environment and intestinal tract of animals. However, the over-abundance of harmful bacteria significantly impacts poultry production [33]. Probiotics can cope with the reduced production performance of aged hens accompanied by the decreased immunity, which has the advantages of no antibiotic residues and antibiotic-resistant bacteria generation [3,34]. This study showed that adding *L. salivarius SNK-6* had no promoting effect on the production performance at any concentration compared with the control group. In contrast, the feed intake and broken-egg rate in the T1 (low-dose) group were significantly higher than those in the Con group. Additionally, a tendency for the FCR to increase was noticed. Correspondingly, the innate immune responses mediated by TLR (e.g., TLR-6, MyD88, NF-κB, IL-1β, IL-10, IL-12) and the major histocompatibility complex (MHC) (e.g., MHC-II, CD80, CD86, CD40, CD40L, and CD28) were significantly activated in cecal tonsil, indicating a lower dose of *L. salivarius* could activate the intestinal mucosal immune responses. Interestingly, no significant difference in the production performance was observed between the T2 (high-dose) and the Con group, corresponding to its relatively modest changes in immune-related gene expression. Both low and high levels of *L. salivarius SNK-6* had the ability to enhance the intestinal protective immune responses.

The cecal tonsil is a crucial gut-associated lymphoid tissue. Many immune cells are distributed in the lamina propria and submucosa of the cecal tonsil, containing a large number of T- and B-lymphocytes and a small number of macrophages and dendritic cells; thus it plays an important role in the intestinal mucosal immune regulation and immune defense function [35,36]. Toll-like receptors (TLRs) play crucial roles in the innate immune system by recognizing pathogen-associated molecular patterns (PAMPs) derived from various microbes, and they can transmit signals into the nucleus through complex actions, activating NF-κB, MAPK, and other signaling pathways to regulate the expression of immune-related genes [37]. MHC-I, MHC-II, CD40, CD80, and CD86 molecules are highly expressed in mature antigen-presenting cells (APCs) such as macrophages and dendritic cells, which provide the necessary costimulatory signals for the activation and proliferation of T- and B-cells and play an immune-regulatory role [38,39,40]. Studies have shown that *L. salivarius* can increase the expression levels of costimulatory molecules (CD40, CD80, CD86) and cytokines (IFN-γ, IL-1β, IL-12, IL-10) in macrophages [41]. In vitro experiments showed that IL-1β, IL-6, and IL-12 in macrophages were upregulated by *L. salivarius* while increasing the mRNA expression levels of IL-1β and IL-12 but had no significant effect on IL-18, IFN-γ, and IL-10 in the cecal tonsil mononuclear cells [42]. Combined with the results of the present trial, we reasonably suggest that *L. salivarius SNK-6* can promote the maturation of the antigen-presenting cells (APC) and subsequently pass the signals to the initial T- and B-cells through recognition of the MHC–peptide complex by the TCR, and costimulatory molecules such as CD28 and CD40L, which promote the activation and proliferation of T- and B-cells and enhance both specific and nonspecific immune response-ability. Additionally, *L. salivarius SNK-6* can improve the intestinal innate immune response by regulating the TLR-mediated signaling pathways [37]. The possible reason for the lack of improvement in the production performance by adding *L. salivarius SNK-6* was probably due to the activation of the immune response, which caused the redistribution of absorbed nutrition, encouraging more nutrition for immunity rather than poultry production.

The intestinal flora is the crucial determinant of immune system maturation and closely correlates with the nature and intensity of intestinal mucosal immune response [43]. Thus, we provided a comprehensive analysis of the cecal microbiome composition to further explore the effects of *L. salivarius SNK-6* supplementation on immune function regulation. The study suggested that the impact of supplying *L. salivarius SNK-6* on the cecal microflora was limited, with no significant difference in the α-diversity and only a trend difference in the β-diversity. Consistent with previous studies, the *Firmicutes*, *Bacteroidetes*, *Proteobacteria*, and *Actinobacteria* were the most dominant phyla in the cecal microbiome of hens, accounting for more than 95% of the total bacteria community [3,44].

LEfSe analysis showed that the control group’s bacteria family Enterobacteriaceae and order RF39 were significantly enriched. At the same time, the genera *Ochrobactrum* and *Eubacterium* were identified as biomarkers to distinguish the two groups by permuted random forest analysis. *Ochrobactrum* (Con: 0.009%, T1: 0.03%, and T2: 0.02%) was a genus with a higher relative abundance in the low and high levels of probiotics groups and concurrently possessed a higher node degree (degree in Con, T1, and T2 groups were 1, 19, and 8, respectively) in the network analysis, indicating that *Ochrobactrum* was more active in the two probiotic-treated groups. Previous work showed that *Ochrobactrum* is a Gram-negative organism closely related to the genus *Brucella* and is typically considered an opportunistic pathogen with low virulence [45]. They include phosphatidylcholine and lipopolysaccharide (LPS) with a lipid A carrying very long-chain fatty acids (VLCFA) [46]. Meanwhile, the genera *Ochrobactrum* was significantly positively correlated with TLR-6, IL-10, MHC-II, and CD40 in cecal tonsils, indicating that it may be involved in enhancing the intestinal immune-related functional indicators in our present study. *Eubacterium* is a potentially beneficial bacterium and is currently identified as the specific strain that produces butyric acid and has glycolysis similar to *Roseburia* and *Faecalibacterium*. It may play a beneficial role like that of *Lactobacillus* and *Bifidobacterium* in regulating intestinal inflammation by producing short-chain fatty acids (SCFA) [47,48]. Studies have revealed that butyric acid exerts anti-inflammatory effects by inhibiting pro-inflammatory cytokines IFN-γ, IL-1β, IL-6, IL-8, and TNF-α, and upregulates anti-inflammatory cytokines IL-10 and TGF-β in an FFAR2/FFAR3 dependent manner [49,50]. However, genera *Eubacterium* (Con: 0.37%, T1: 0.20%, and T2: 0.23%) had a higher relative abundance in the non-probiotics control group. Order *RF39* is a potentially beneficial bacterium for health associated with healthy dietary patterns, but negatively correlated with blood triglycerides in many studies [51,52,53]. *Enterobacteriaceae* is a large family of Gram-negative bacteria including much beneficial commensal microbiota and a wide range of disease-causing pathogens (e.g., *Salmonella, Escherichia coli, Klebsiella*, and *Shigella*) [54]. Only one ASV was classified as family *Enterobacteriaceae* in this trial; thus, it should be judged with caution as to which genus or strain plays an essential role in regulating the immune response. Taken together, in the lower level of the *L. salivarius SNK-6* group, the relative abundance of genera *Ochrobactrum* was markedly increased, which may enhance gut immune function. In contrast, the relative abundance of genera *Eubacterium* and order *RF39* decreased, inhibiting the intestinal inflammation response.

Interestingly, the high dose of *L. salivarius SNK-6* displayed relatively weaker immune responses than the lower level of probiotic treatment. The bacteria family *Enterobacteriaceae*, genera *Megamonas*, species *Bacteroides barnesiae*, and *Bacteroides Plebeius* were significantly enriched in the non-probiotic control group. In contrast, the order *SHA-98* and species *Desulfovibrio D168* were increased considerably in the high-concentration group by LEfSe analysis. The genera *Bacteroides*, *Alistipes*, and *Megamonas,* were identified as the critical bacteria to distinguish the two groups by permuted random forest analysis. We found that explicit enrichment of bacteria species *Desulfovibrio D168* and order *SHA-98* in the high-dose group was significantly negatively correlated with the expression of IL-12, suggesting that the taxa of these two bacteria may be related to the reduction in the intestinal immune response. Order *SHA-98* belongs to the class *Clostridia*. Some studies have been associated with healthier gut microbial metabolism and reduced inflammatory or allergic reactions, which were compatible with our results [55,56]. *Megamonas* (con: 1.16%, T1: 1.19%, and T2: 0.26%) is a genus that belongs to obligate anaerobes and can ferment glucose into acetate and propionate with decreased relative abundance in the high-dose probiotics group [57,58]. Meanwhile, research has shown that the genus *Megamonas* is related to inflammatory reactions [59]. The genera *Bacteroides* (Con: 14.88%, T1: 11.83%, and T2: 9.98%) is the dominant bacteria in cecum, which are mainly involved in decomposing complex molecules into simpler compounds [44,60]. Genera *Bacteroides* was positively correlated with weight gain and growth performance and could inhibit *Clostridium perfringens* by its metabolites in poultry research [61,62]. We found that the relative abundance of the genera *Ochrobactrum*, which was positively related to immunity function, decreased in the high-dose probiotics group compared to the low-dose group but was higher than that of the control group. In summary, we suggest that the relatively modest immune response in the high-dose probiotics group could be due to the relative abundance of bacteria related to the pro-inflammatory being lower than that of the low-dose probiotics group. High-dose *L. salivarius SNK-6* enriches some bacteria associated with suppressing the immune response, resulting in milder immune responses in the cecal tonsils of hens.

## 5. Conclusions

Overall, supplementary *L. salivarius SNK-6* at 1.0 × 10^6^ CFU/g of feed induced stronger intestinal innate immune responses than the high-dose group (1.0 × 10^7^ CFU/g), possibly due to different factors influencing the microbial community composition of the cecum. However, the lower dose of *L. salivarius SNK-6* also caused a reduced production performance related to the activation of the immune response, which needs to be weighed up in practical applications.

## Figures and Tables

**Figure 1 microorganisms-10-01469-f001:**
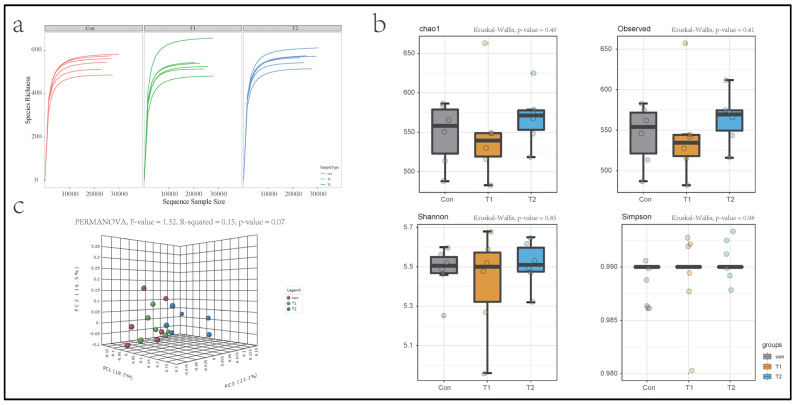
The overall description of the gut microbiota in different groups. (**a**) The observed species dilution curve evaluated the rationality of the sequencing depth. (**b**) The α-diversity of microbial communities is shown as richness and evenness. (**c**) The principal coordinate analysis of the ileal microbial community.

**Figure 2 microorganisms-10-01469-f002:**
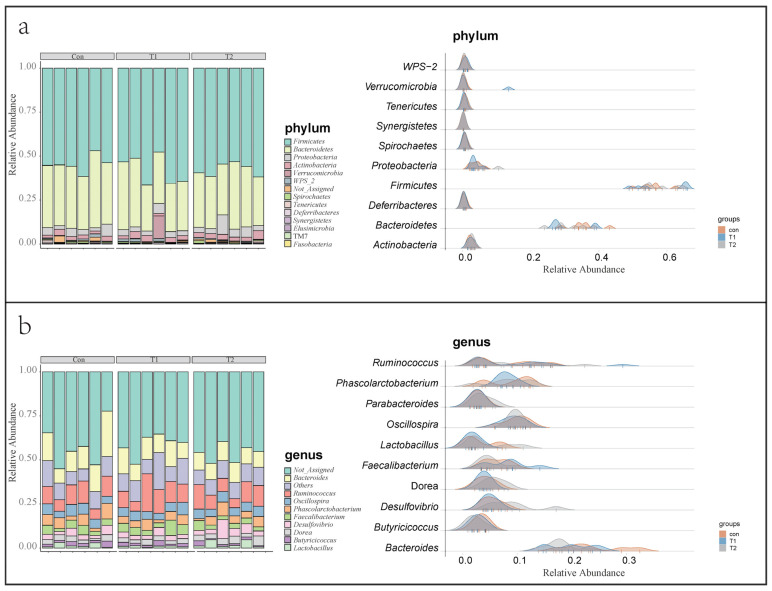
The microbial community structure of cecum. (**a**) Left: stacked bar chart of microbial composition at the phylum level; Right: ridges chart of the microbiota at the phylum level. (**b**) Left: stacked bar chart of microbial composition at the genus level; Right: ridges chart of the microbiota at the genus level.

**Figure 3 microorganisms-10-01469-f003:**
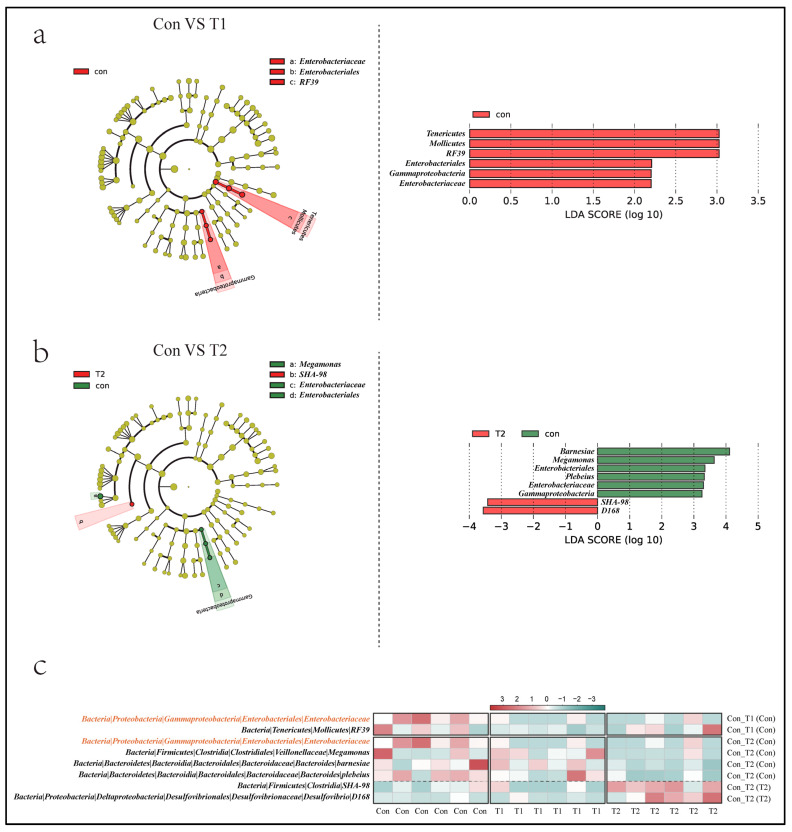
The significantly differentially abundant bacteria taxonomy identified by LEfSe analysis. (**a**) Cladogram plot (**left**) and Histogram of LDA value (**right**) between the Con and T1 group. (**b**) Cladogram plot (**left**) and Histogram of LDA value (**right**) between the Con and T2 group. (**c**) The heat map of the relative abundance of landmark microorganisms identified in the LEfSe in each group, the darker red represents the higher degree of enrichment, and the darker green represents the lower degree of enrichment.

**Figure 4 microorganisms-10-01469-f004:**
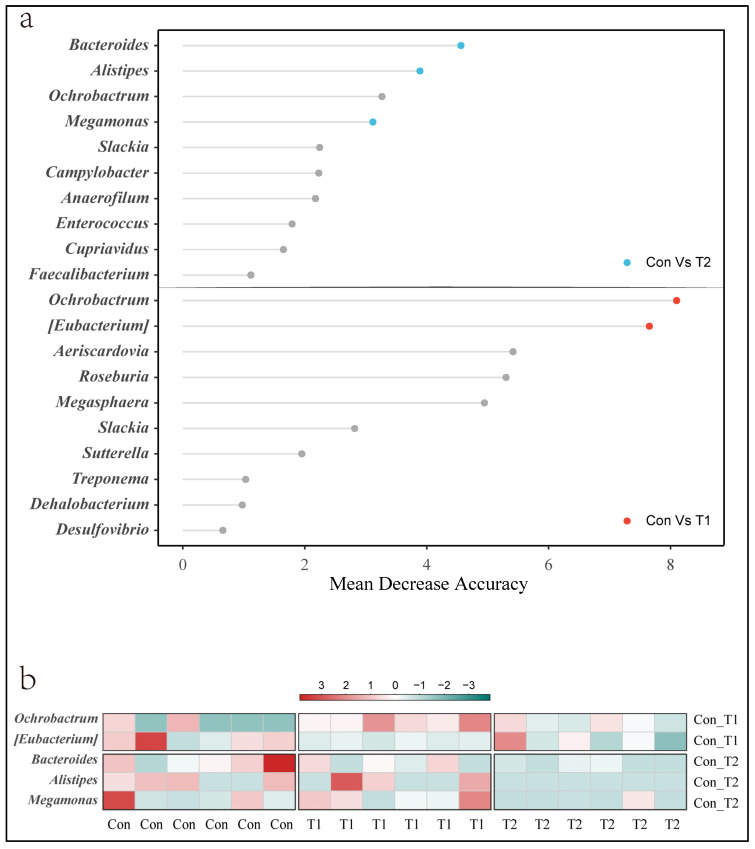
The machine-learning classification based on genera in cecal microbial using random forest algorithms. (**a**) The top 10 genera according to relative importance by the mean decrease accuracy of “Con vs. T1” and “Con vs. T2” (the number of permutations was 1000 times). The red circle represents the genera that were confirmed and *p* < 0.05 simultaneously in “Con vs. T1”. The blue circle represents the genera that were confirmed and *p* < 0.05 simultaneously in “Con vs. T2”. (**b**) The heat map of the relative abundance of the genera identified through random forest.

**Figure 5 microorganisms-10-01469-f005:**
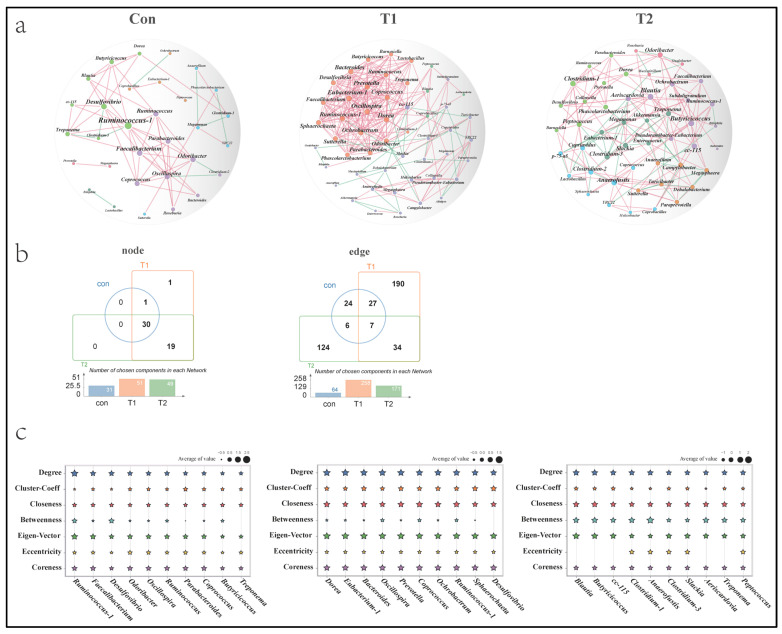
The co-expression correlation network analysis of the microbial community. (**a**) Each node represents a genus, and each node’s size represents the nodal degree. The node with the same color represents the same module. The red edges represent a positive correlation, and the blue edges represent a negative correlation. (**b**) Venn diagrams of the nodes and edges in each netwo rk (**c**) According to the nodal degree ranking, network topological properties were drawn across each treatment. Different star colors represent different network topological properties.

**Figure 6 microorganisms-10-01469-f006:**
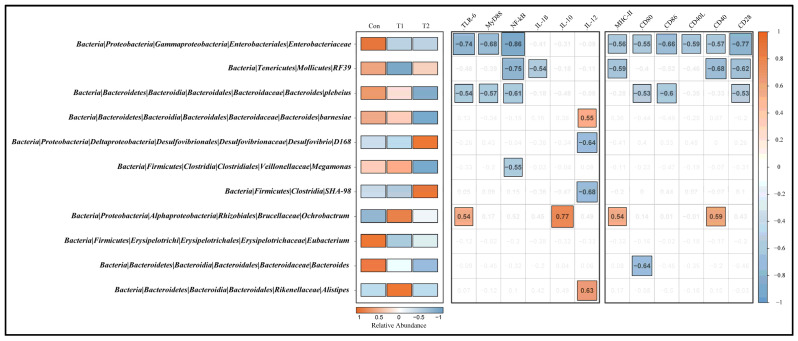
The correlation analysis was performed between the signature microbiota (cecal) and differential immune-related gene expression (cecal tonsil). From left to right: genus names; the relative abundance of the genera in different groups. The darker the orange, the higher the relative abundance, while the darker the blue, the lower the relative abundance. The correlation between gene expression and signature microbiota. The orange squares represent positive correlation, blue squares represent negative correlation, while blank squares represent no significant difference (n.s.).

**Table 1 microorganisms-10-01469-t001:** The ingredients and nutrient composition of the basal diet.

Ingredients	Percent (%)	Nutrient Levels	Content
Corn (CP 8.3%)	64.00	ME (MJ/Kg)	11.16
Soybean meal (CP 44.0%)	19.80	CP (%)	15.00
Soybean oil	0.70	CF (%)	2.33
Wheat bran (CP 14.3%)	4.11	Met (%)	0.33
Limestone	9.50	Lys (%)	0.77
Calcium hydrogen phosphate	1.00	Met + Cys (%)	0.67
Sodium chloride	0.30	Trp (%)	0.17
DL-Methionine (98%)	0.10	Thr (%)	0.59
L-Lysine HCL (78%)	0.07	Calcium (%)	3.72
Vitamin premix ^a^	0.03	Total *p* (%)	0.52
Mineral premix ^b^	0.20	Available *p* (%)	0.34
Choline chloride (50%)	0.15		
Phytase	0.02		
NSP enzymes	0.02		
Total	100.00		

^a^ Supplied per kilogram of diet: vitamin A, 13,500 IU; vitamin D3, 4500 IU; vitamin E, 75 IU; vitamin K3, 3.6 mg; vitamin B1, 3.0 mg; vitamin B2, 9.24 mg; vitamin B6, 6.0 mg; nicotinic acid, 66 mg; pantothenic acid, 16.8 mg; biotin, 0.54 mg; folic acid, 2.10 mg; vitamin B12, 0.03 mg; choline, 675 mg; ^b^ Mineral premix provided per kilogram of complete diet: iron, 80 mg; copper, 10 mg; manganese, 100 mg; zinc, 100 mg; iodine, 0.35 mg; selenium, 0.30 mg.

## Data Availability

The microbial raw sequencing data were uploaded to the NCBI Sequence Read Archive database (SRA accession: PRJNA792225).

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
