# Peer review of "Lactobacillus salivarius SNK-6 Activates Intestinal Mucosal Immune System by Regulating Cecal Microbial Community Structure in Laying Hens"

_microorganisms, 2022, doi:10.3390/microorganisms10071469_

Round 1

Reviewer 1 Report

I would like to congratulate the author for the well written manuscript and the good job on the experiments. The study provided insights for the mechanisms of how L. salivarius SNK-6 modulated the intestinal microbiota and immune response in laying hens. Also, relatively optimum supplemented dose for this probiotic strain was also indicated by the author.

There are some parts that require moderate revision:

1.      Please thoroughly check the grammar, language, formats, and the punctuations used in this manuscript. For example, Latin terms such as in vivo and ad libitum should be italicized.

2.      For feed ingredients used in this study, please provide a reference for the feeding standard NY/T-33-2004 in the reference section.

3.      The average temperature in the chicken house during the trial period should also be provided in the materials and methods section.

4.      In Table 1, please supplemented with analyzed value for the gross composition in feed, including CP, EE, DM, ME, CF (or ADF and NDF). Also, in the materials and methods section, please provide feed analysis methods to cooperate with the analyzed results.

Author Response

Dear editor and reviewers:

Thank you for your positive response to our work and the kind advice. We greatly appreciate your constructive comments that have helped us improve our paper. We have endeavored to incorporate the feedback and revised our manuscript accordingly. We have checked the references in the manuscript and all changes in the manuscript have been made using the "Track Changes" mode and highlighted. The itemized response is as follows:

Q1: Please thoroughly check the grammar, language, formats, and punctuations used in this manuscript. For example, Latin terms such as in vivo and ad libitum should be italicized.

A1: Thank you for your comment and kind advice. The full text's grammar, language, and format were thoroughly checked. In particular, in vitro (line 63, 365) and ad libitum (line 97) have been changed to italics and highlighted in the revised text.

Q2: For feed ingredients used in this study, please provide a reference for the feeding standard NY/T-33-2004 in the reference section.

A2: Thank you for your comment and kind advice. The reference for NY/T-33-2004 feed standard (Line 91,535) has been supplemented.

Q3: The average temperature in the chicken house during the trial period should also be provided in the materials and methods section.

A3: Thank you for your comment and kind advice. The materials and methods section added the chicken house's environment and ambient temperature during the test (line 94 -- 96).

Q4: In Table 1, please supplement the analyzed value for the gross composition in feed, including CP, EE, DM, ME, CF (or ADF and NDF). Also, in the materials and methods section, please provide feed analysis methods to cooperate with the analyzed results.

A4: Thank you for your comment and kind advice. Unfortunately, in this experiment, we used calculated values instead of measured values for all the nutrients in the feed since the only variable in the treatment and control group was whether to supply L. salivarius SNK-6 or not. However, we have seriously considered your suggestion, and we will pay special attention to this point in future research and detect the corresponding gross composition of feed materials. Thanks for your reminder. When we asked the nutrition specialist to review our article, we found that there might be some minor problems in the calculation of nutrients, which have been revised here. Please refer to table1 for details.

Reviewer 2 Report

The manuscrip is well written an contains many important information about immunoregulatory effect and intestitial microbiota content in laying hens after Lactobacillus salivarius SNK6 application.The Authors confirmed that high-dose L. salivarius SNK-6 enriches some bacteria associated with sup-pressing the immune response, resulting in milder immune responses in the cecal tonsils of hens. However, lower dosis of this probiotic bacteria has an influence on immunomostimulation effect.

Please also read the text carefully and correct all typing errors.

In my opinion the article needs minor revision and could be acceptable for publication after this correction.

All comments has been included in the text of manuscript.

Author Response

Dear editor and reviewers:

Thank you for your positive response to our work and the kind advice. We greatly appreciate your constructive comments that have helped us improve our paper. We have endeavored to incorporate the feedback and revised our manuscript accordingly. We have checked the references in the manuscript and all changes in the manuscript have been made using the "Track Changes" mode and highlighted. The itemized response is as follows:

Q1: Please add the information about the breeding system.

A1: Thank you for your comment and kind advice. We supplemented the information related to the breeding system in Abstract (Line 16-17) and Materials and Methods (Line 94-97). In the Materials and Methods section, detailed feeding environment information was added.

Q2: Please add in this part of the text what kind of immunoregulatory effect is caused by this probiotic strain after, e.g.…

A2: Thank you for your comment and kind advice.

We have carefully read this sentence and don't think it plays a crucial role in summarizing their implication in the immune system, even misleading readers, so we removed the redundant description here.

In the Introduction, we systematically reviewed the regulatory effect of Lactobacillus salivarius on the immune system, which is divided into three parts:

  1. When the host or cell is stimulated by LPS and pathogenic microorganisms (e.g., bacteria and viruses), Lactobacillus salivarius often exerts anti-inflammatory and anti-infection effects by producing anti-inflammatory responses (promoting the production of anti-inflammatory cytokines or antibodies). (Line56-59).
  2. In the absence of an infection, macrophages (other immune cells such as dendritic cells) can be stimulated to express cytokines or produce substances such as antimicrobial peptides. T cells can be promoted to produce cytokines or B cells to produce antibodies. (Line60-63).
  3. Lactobacillus salivarius can promote Th1 cell differentiation in vitro and thus participate in immunomodulatory responses (Line63-65).

Q3: Suggest adding some information about the mechanism of influence on intestinal microflora content.

A3: Thank you for your comment and kind advice. The potential impact mechanism on gut microflora has been added in the Introduction section. (Line 53-56)

Q4: Did the authors resuspend in water or something like that this freeze-dried powder containing lactobacillus salivarius strain? If yes, please add this information. Please explain if the authors used this powder directly in the feed about the proportion of used powder and the total feed amount.

A4: Thank you for your comment and kind advice. We didn't resuspend in water; instead, we directly added the lyophilized bacteria powder into the feed. Professor Yan Huaxiang provided the lyophilized bacteria powder. Then we used the agar plate method to detect the number of bacteria and expressed it as cfu/g. The number of viable bacteria in the lyophilized bacteria powder was confirmed to be 1.0×1011CFU/g (Line 73-75).

Q5: Please add the information about the region of country, season of the year.

A5: Thank you for your comment and kind advice. This information has been added in the Materials and Methods section. Please see Line 94-96.